# The Use of a Dental Storybook as a Dental Anxiety Reduction Medium among Pediatric Patients: A Randomized Controlled Clinical Trial

**DOI:** 10.3390/children9030328

**Published:** 2022-03-01

**Authors:** Alrouh M. Alsaadoon, Ayman M. Sulimany, Hebah M. Hamdan, Ebtissam Z. Murshid

**Affiliations:** 1Department of Pediatric Dentistry and Orthodontics, College of Dentistry, King Saud University, Riyadh 11545, Saudi Arabia; asulimany@ksu.edu.sa (A.M.S.); emurshid@ksu.edu.sa (E.Z.M.); 2Department of Periodontics and Community Dentistry, College of Dentistry, King Saud University, Riyadh 11545, Saudi Arabia; hhamdan1@ksu.edu.sa

**Keywords:** dental fear, children’s fear survey schedule-dental subscale, Venham clinical anxiety scale, Frankl behavior rating scale, preparatory information, behavior management

## Abstract

This randomized clinical trial aimed to evaluate the effectiveness of a specially designed dental storybook in reducing dental anxiety among children. Eighty-eight children (6–8 years old) were randomly divided into two groups: the intervention group (received the storybook) and the control group (did not receive the storybook). Three dental visits (screening, examination and cleaning, and treatment) were provided for each child. Anxiety was assessed following each visit using the Children’s Fear Survey Schedule-Dental Subscale (CFSS-DS) and the Venham clinical anxiety scale (VCAS). The behavior was assessed using the Frankl’s Behavior Rating Scale (FBRS). The intervention group showed significantly lower anxiety and more cooperative behavior during treatment than the control group (*p* < 0.0001). The intervention group showed a significant decrease in anxiety scores and more cooperative behavior across time according to the CFSS-DS (*p* = 0.001) and Frankl behavior scale OR = 3.22, 95% CI 1.18–8.76. Multivariate models found that using the storybook was a significant independent factor in reducing anxiety and improving behavior after controlling for sex, previous visits, family income, and mother’s education. In conclusion, the dental storybook can decrease children’s dental anxiety and improve their behavior during dental treatment.

## 1. Introduction

Dental anxiety is a significant social problem among children. It can be defined as a non-specific feeling of apprehension requiring no prior experience of the situation anticipated [1,2]. The etiology of dental anxiety is multifactorial. It could be due to exogenous factors, endogenous factors, or a combination of both [3]. External (exogenous) sources include direct or indirect negative vicarious experiences. Endogenous sources are personality traits, cognitive abilities, and heritability [4]. Several studies have confirmed that the main cause of dental anxiety is either pain or fear related to pain [5,6,7]. Patients who are more sensitive to pain are likely to have a higher level of anxiety [7].

Multiple studies have estimated the prevalence of dental anxiety among children. In 1997, an epidemiological investigation was conducted in eight European countries and found that 35% of 5 year-old children exhibited significant fear before going to the dentist [8]. In 2004, a study in Washington’s private pediatric dentistry practices found that 20% of 1–13 year-old children had dental fear, and 21% presented negative behaviors during treatment [9]. More recently, in Saudi Arabia, it was found that 57% of 6–12-year-old children were highly anxious during their dental visits [10].

Anxiety in dental patients generally manifests physiologically, including symptoms in terms of fright response. In addition to the physiological reactions, children can have behavioral and cognitive reactions [11]. Therefore, it is of the utmost importance to manage patients with dental anxiety or dental fear in a manner that complements their conflicts. Various behavior management techniques have been proposed by the American Academy of Pediatric Dentistry (AAPD) to manage the uncooperative behavior of pediatric dental patients. These techniques range from basic behavioral management approaches (Tell-Show-Do, positive reinforcement, distraction, and voice control) to more advanced and invasive approaches, such as protective stabilization, sedation, and general anesthesia [12].

A noteworthy approach to behavioral management is based on Bandura’s social learning theory. It posits that people learn by observation, imitation, and modeling [13]. This includes giving preparatory information regarding the procedure to the pediatric patient, which can decrease the discomfort and pain perception [14]. Moreover, self-regulation theory (SRT) can explain the viability of preparatory information. An important aspect of self-regulation theory assumes that knowing what will happen makes the situation less stressful [15]. Given both theories, exposing children to positive information regarding dentistry, such as images or storybooks of enjoyable dental activities, can reassure them and psychologically prepare them for their dental visits [16].

The use of stories in healthcare has several goals: educating patients and their families, promoting specific traits, and enhancing certain behaviors [17]. Several studies have been conducted to determine the effect of preparatory information on patient anxiety, specifically storybooks. This was elaborated in some medical studies that evaluated the effectiveness of storybooks on reducing anxiety before surgeries. Most studies revealed a reduction in anxiety before surgeries [18,19,20].

Dental studies on the effectiveness of preparatory information before dental visits yielded mixed results. Fox and Newton exposed children to positive images of dental-related materials, and they reported a reduction in dental anxiety among British children [16]. Similarly, Moura et al. evaluated anxiety levels among children before and after showing them an audio-visual book before their dental appointments. The book led to a significant reduction in the anxiety levels among these children [21]. Moreover, previous studies have also investigated the effectiveness of using dental stories as a preparatory tool before dental visits in children with autistic spectrum disorder. The results suggested that dental stories had a positive impact on the children’s behavior [22,23].

In contrast, having previous dental care information showed no significant effect on dental anxiety among Nigerian children [24]. Similarly, Olumide et al. tested whether viewing leaflets entailing positive dental information influenced the children’s dental anxiety. The authors did not report a reduction in anticipatory anxiety following the provision of preparatory information [14].

The evidence regarding the effectiveness of preparatory information in the dental field is inconclusive. Moreover, none of the published studies have yet evaluated the effectiveness of a dental storybook in reducing dental anxiety over multiple visits. Therefore, this study evaluated the effectiveness of a specially designed dental storybook in reducing dental anxiety and improving behavior among children during examination and treatment plan visits, followed by a restorative dental visit.

## 2. Material and Methods

### 2.1. Study Design and Ethical Approval

This was a two-arm parallel, single-blind, randomized controlled trial (RCT) conducted at the pediatric dental clinics of the Dental University Hospital at King Saud University, Riyadh, Saudi Arabia. Ethical approval was obtained from the Institutional Review Board (IRB) at the College of Medicine, King Saud University under research number E-18-3190 in 20 March 2019. The study protocol received institutional approval from the Ethics Committee of the College of Dentistry Research Center (CDRC) of the King Saud University in Riyadh, Saudi Arabia (number PR 0104). This study was registered at the ISRCTN with study ID ISRCTN44193972.

The study was performed in accordance with the Declaration of Helsinki and followed CONSORT guidelines. All children and their parents received written and verbal information about the procedure before inclusion. Children gave assent prior to participation, and parental informed written consent was also obtained.

### 2.2. Participant Screening and Eligibility Assessment

Children aged 6–8 years who met the inclusion criteria were included in this randomized clinical trial.

The inclusion criteria were: medically fit children with ASA I (normal healthy patients) according to the American Society of Anesthesiologists Classification [25]; children/parents able to read and understand Arabic; and in need of restorative treatment (occlusal fillings) that would require local anesthesia in the upper arch.

The exclusion criteria were children with special needs; children with a complete audio-visual impairment; children with learning difficulties or mental retardation; children who were not Arabic speakers; previous treatment with nitrous, sedation, or general anesthesia; conditions requiring emergency dental treatment (abscess, draining sinus, cellulitis); need for pharmacological management to cooperate; and known dental phobia.

### 2.3. Sample Size Calculation

The power of the sample was calculated using the G power sample power calculator (University of Kiel, Kiel, Germany). A sample size of 88 was needed for an effect size of 0.25 and a power of 0.95 for a two-sided normal distribution with the two groups. Anticipating a possible attrition rate of 20%, an estimated sample size of 105 was required.

### 2.4. Randomization and Blinding

All the eligible children were randomized to one of the two groups using the block randomization method: an intervention group (who received the storybook) and a control group (who did not receive the storybook). The required sample size of 105 was divided into seven blocks with 15 subjects in each block. A block of 15 two-digit random numbers was generated from which odd/even random numbers were allotted to the intervention and control groups. An independent trial investigator performed allocation concealment with sequentially numbered, opaque, and sealed envelopes (not measuring the study’s outcomes). The allocation ratio was intended to be equal on both sides. The main investigator (outcome assessor A.M.A.) was blinded to the group allocations.

### 2.5. Study Design

The study comprised three visits: screening, examination, and restorative treatment, which were performed by a senior pediatric dentistry resident (A.M.A.).

#### 2.5.1. Screening Visit

The children were selected during this visit. A simple oral examination was performed to assess whether the child was suitable for the study. Suitable children’s parents/guardians were given consent. Demographical information was collected from the parents/guardians.

At the end of the visit, the child’s baseline anxiety was assessed using the Children Fear Survey Schedule-Dental Subscale (CFSS-DS) and the Venham clinical anxiety scale (VCAS). Children’s behavior was assessed using the Frankl behavior rating scale. An independent investigator then performed subject allocation to either group.

The investigator was further responsible for distributing the storybook to the intervention group and giving instructions to the parents to read the book to their children twice (when they received the book and one day before their examination dental visit) to prepare the children for their next dental visits. The independent investigator was not involved in assessing the outcome measures.

#### 2.5.2. Examination Visit

A complete medical and dental history was obtained from parents/guardians. Children underwent both extraoral and intraoral examinations, prevention measures, radiographs if necessary, and fluoride therapy. A treatment plan was formulated for each child.

During this visit, the tell-show-do psychological behavior management technique was used to introduce the child to dental procedures. At the end of the visit, dental anxiety levels were assessed using the previously used scales, the Children Fear Survey Schedule-Dental Subscale (CFSS-DS) and the Venham clinical anxiety scale (VCAS). Children’s behavioral ratings were assessed using the Frankl behavior rating scale.

#### 2.5.3. Treatment Visit

In this visit, children received occlusal composite restorations (Filtek Supreme XTE (3M ESPE, St. Paul, MN, USA)) that required the administration of local anesthesia (2% lidocaine with 1:100,000 epinephrine) in the upper arch. All restorations were performed under rubber dam isolation. At the end of the visit, both anxiety levels and behavior were assessed using the same scales: CFSS-DS, VCAS, and Frankl behavior rating scale.

The interval between the visits was one week. Parental presence in the dental office was allowed in all visits with basic behavior guidance. At the end of the study, children who needed additional treatment were scheduled with the same dentist.

### 2.6. The Storybook Intervention

The storybook was designed to prepare children for their dental visits. It utilizes a specific color scheme intertwined with cartoon characters to assist in building the narrative. The book describes and explains the various stages and peripherals attached to the first dental visit, including an examination (prophylaxis and topical fluoride). The story is written in simple Arabic and describes the waiting area, the dentist’s and dental assistance’s roles, the instruments and their uses, and the clinic. The goal is to familiarize children with the dental visits. The design of the book and language were reviewed and modified by experts in Arabic language and childhood education before it was approved and registered at the Ministry of Culture and Information of Saudi Arabia (registration number ISBN:978-603-02-0122-8) (Figure 1).

### 2.7. Outcome Measures

The children’s anxiety and behavior were assessed at the end of each visit. The anxiety levels were assessed using CFSS-DS and VCAS. CFSS-DS is a 5-point Likert scale. It comprises 15 items involving unique characteristics of dental care, and responses ranging from 1 (in which a person is not afraid at all) to 5 (being very afraid) [26]. The Arabic version of the CFSS-DS showed high reliability and validity among Arabic-speaking children [27]. The VCAS has also demonstrated high reliability and validity [28]. It is quick, easy to use, and widely used by many dentists. It can be easily incorporated into clinical situations and research structures [24,29,30]. It categorizes children into six groups based on their behavior (0–5); 0 = relaxed, 1 = uneasy, 2 = tense, 3 = reluctant, 4 = interference, and 5 = out of contact.

Children’s behavior was assessed using the Frankl Behavior Rating Scale with confirmed validity [31]. It is one of the most reliable tools for rating the behavior of children in a dental setting and has shown 100% reliability [32,33,34]. It allows sorting of patients into definitely negative, negative, positive, and definitely positive categories [35].

### 2.8. Intra-Examiner Reliability

The main investigator (A.M.A.) was trained and calibrated in assessing anxiety and behavior. For validation purposes, the intra-examiner reliability of the VCAS and the Frankl Behavior Rating Scale was tested in a pilot study of ten videotaped patients. The intra-examiner reliability scores of the VCAS and the Frankl Behavior Rating Scale after 2 weeks were 0.92 and 0.89, respectively.

### 2.9. Statistical Analysis

The normality check was performed using the Shapiro–Wilk test. The normality assumption for the CFSS-DS scores was not violated at the screening visit, and thus the independent sample *t*-test was used to test for differences in the screening visit scores between the intervention and control groups. The Wilcoxon rank-sum test was used to test for differences in CFSS-DS scores in the intervention group between the examination and treatment visits.

To simplify statistical analysis, the VCAS scores were re-categorized into 2 scores: 0 and 1 (indicate positive behavior) and 2–5 (indicate negative behavior). The Frankl behavior rating scores were categorized into negative (ratings 1 and 2) and positive (ratings 3 and 4). A Chi-squared test was used to assess the differences in VCAS and Frankl scores between the groups at each visit.

Mixed-effects negative binomial regression with subject-level random effects was performed separately for each group to assess the changes in CFSS-DS scores with time in each group. For Venham and Frankl scores, mixed-effects logistic regression models with subject-level random effects were used separately for each group to assess the changes in Venham and Frankl scores over time in each group. Multivariate, mixed-effects negative binomial regression and logistic regression models were developed to assess the effect of the intervention in terms of CFSS-DS, Venham, and Frankl scores after controlling for visits, age, sex, previous dental visits, family income, and mother’s education. These factors were chosen based on previous studies related to pediatric dental anxiety.

The significance level for all tests was set to *p* ≤ 0.05, and all statistical analyses were performed using SAS software (Statistical Analysis Software 9.4, SAS Institute Inc., Cary, NC, USA).

## 3. Results

This study was conducted from January 2019 to March 2020. Figure 2 is a flowchart of the children who participated during each trial phase: enrollment, allocation, follow-up, and data analysis.

### 3.1. General Characteristics of the Children and Their Parents

Table 1 illustrates the demographic details of the population. The sample comprised 88 children (41 males, 46.6%; 47 females, 53.4%) between 6 and 8 years old (mean age ± SD: 7.08 ± 0.76 years) who received the storybook (*n* = 43, 48.9%) or did not (*n* = 45, 51.1%). The participating children were mainly Saudis (*n* = 79, 89.8%), and most participating families were middle-class socio-economically (*n* = 47, 53.4%). Many parents had a bachelor’s degree (*n* = 38, 43.2% for fathers; *n* = 42, 47.7% for mothers). Children were predominantly accompanied by their fathers, who were responsible for reading the book to their children (*n* = 46, 52.3%). Parents’ responses indicated that 43 participating children had previous dental visits (48.9%), whereas 45 of them had not (51.1%). In general, there were no significant differences between the two groups with regard to their background characteristics.

### 3.2. Anxiety and Behavioral Measurements

#### 3.2.1. CFSS-DS

There were no significant differences between the intervention and control groups in CFSS-DS scores in the screening visit when the dental anxiety of the two groups was compared at each of the three visits (*p* = 0.694). However, there were significantly lower CFSS-DS scores among the intervention group during both the examination visit (*p* = 0.049) and the treatment visit (*p* < 0.0001) as compared with the control group (Table 2).

Mixed-effects negative binomial regression with subject-level random effects was performed separately for each group to test the changes in CFSS-DS scores over time for each group. The results suggest that there was a significant increase in CFSS-DS scores in the control group from the screening visit to the treatment visit (*p* = 0.001). In contrast, there was a significant decrease in CFSS-DS scores from the screening visit to the treatment visit in the intervention group (*p* = 0.001) (Table 3).

A multivariate, mixed-effects negative binomial regression model with subject-level random effects was developed to assess the effect of the dental storybook in CFSS-DS after controlling for the type of dental visit, age, sex, previous dental visits, family income, and maternal education. The intervention group had significantly lower CFSS-DS scores than the control group (*p* = 0.002). Moreover, girls had significantly higher CFSS-DS scores than boys (*p* = 0.02) (Table 4). The type of dental visit, age of the child, previous dental visit, family income, and maternal education were not significantly associated with CFSS-DS scores.

#### 3.2.2. VCAS Scores

When the Venham scores were compared between the groups, it was observed that there were no significant differences between the intervention group and the control group in the screening visit or the examination visit. However, there were statistically significant differences between the groups in the treatment visit, where only 24.3% of the intervention group had negative scores compared with 75.7% in the control group (*p* < 0.0001) (Table 5).

Mixed-effect logistic regression with subject-level random effects was performed separately for each group to test the changes in Venham scores over time for each group. In the intervention group, the results suggest that the examination and treatment visits had higher odds of having positive Venham scores than the screening visit. However, these effects were not significant at alpha = 0.05.

In the control group, the results suggest that the examination and treatment visits had lower odds of having a positive Venham score compared with the screening visit. However, only the treatment visit was significant with an odds ratio of 0.10 and a 95% CI of 0.03–0.38 (Table 6).

A multivariate mixed-effects logistic regression model with subject-level random effects was developed to assess the effect of the dental storybook in Venham scale after controlling for the type of dental visit, age, sex, previous dental visits, family income, and maternal education. The results show that the intervention group had significantly higher odds of having positive Venham scores than the control group (OR = 2.34, 95% CI 1.21–4.55).

Moreover, previous dental visits and maternal education were significantly associated with dental anxiety. Children with previous dental visits had significantly higher odds of having positive Venham scores than those who did not (OR = 2.60, 95% CI 1.29–5.23). The children of mothers with bachelor’s degrees had significantly lower odds of having positive Venham scores compared with those with high school educations or less (OR = 0.35, 95% CI 0.16–0.77). The type of dental visit and the child’s age were not significantly associated with Venham scores (Table 7).

#### 3.2.3. Frankl Scores

There were no significant differences between the intervention group and control group in the screening visit when the Frankl scores between the groups were compared. However, there were significant differences between the groups in the examination visit (*p* = 0.039) and treatment visit: only 22.9% of the intervention group had a negative score compared with 77.1% in the control group (*p* < 0.0001) (Table 8).

Mixed-effects logistic regression with subject-level random effects was performed separately for each group to test each group’s changes in Frankl levels over time. In the intervention group, the results suggest that the examination visit and treatment visit were related to significantly higher odds of having positive Frankl scores compared with the screening visit (OR = 3.79, 95% CI 1.35–10.68, and OR = 3.22, 95% CI 1.18–8.76, respectively). In the control group, the results suggest that the examination and treatment visits were both related to lower odds of having positive Frankl scores compared to the screening visit. However, only the treatment visit was significant with an odds ratio of 0.18 and a 95% CI of 0.07–0.47 (Table 9).

A multivariate mixed-effects logistic regression model with subject-level random effects was developed to assess the effect of the dental storybook on the Frankl score after controlling for the type of dental visit, age, sex, previous dental visits, family income, and maternal education. The intervention group had significantly higher odds of positive Frankl scores than the control group (OR = 2.23, 95% CI 1.13–4.39).

Previous dental visits had borderline significance. Children with previous dental visits had higher odds of having positive Frankl scores compared with those who did not (OR = 2.01, 95% CI 0.99–4.08). Maternal education had a significant association with dental anxiety. Children of mothers with bachelor’s degrees had significantly lower odds of positive Frankl scores compared to those with high school educations or less (OR = 0.36, 95% CI 0.16–0.79). The type of dental visit, age of the child, and family income were not significantly associated with Frankl scores (Table 10).

## 4. Discussion

Understanding dental fear and anxiety in young children is critical to minimizing their fear and anxiety pre- and peri-operatively and in managing their behavior [36]. Among the vast behavior management options available today, this study explored a psychological approach to the management of children’s dental anxiety based on social learning theory [13]. Children were exposed to a specially designed storybook that showed dentistry in a positive way before their dental visits to assess the effectiveness of this approach in reducing dental anxiety among children.

The results found a significant difference in the overall anxiety levels between the intervention and control groups. Children in the intervention group showed a noticeable reduction in their anxiety levels during the treatment visits compared with the control group. This study’s findings agree with other studies conducted to evaluate the anxiety levels of children during dental visits. Aminabadi et al. used a similar approach and suggested that storytelling effectively ensures cognitive development and growth among children in both modeling and procedural forms. Their study was performed to determine the impact of asking children to listen to a relevant story (with pictures) read to them by one of their parents. Their results indicated significant decreases in situational anxiety and perception of pain during dental treatment [37].

Similar findings were reported by Moura et al. via a different technique. They evaluated anxiety levels among children before and after applying a playful tool (audio-visual book) before the dental appointments. Exposure to this tool led to significantly reduced anxiety among those children [21]. Moreover, Elicherla et al. compared the effectiveness of a smartphone application (Little Lovely Dentist) with the tell-show-do (TSD) technique in reducing the dental anxiety during first visits. They found that children educated using the application prior to their dental visits had significantly lower anxiety levels compared with the TSD group [38].

On the other hand, Olumide et al. investigated whether children’s dental anxiety could be influenced by viewing leaflets containing positive dental information. Following the presentation of the leaflets, the authors found no reduction in anticipatory anxiety level [14]. This may have been related to the method through which the anxiety was measured—i.e., they only used a self-reported measure. The differences between Olumide’s et al. study and the current study may be attributed to the methods of delivering information. Storybooks can offer more in-depth information than leaflets, and children can get useful information in an easy and entertaining way. The frequency of reading can also impact the children’s anxiety. In Olumide’s et al. study, leaflets were only shown to the children once, whereas in this study the storybook was read twice. This helped solidify the information and led to a notable behavioral change in the dental clinic.

Olumide et al. corroborated Huntington et al., who evaluated the effectiveness of an online family-centered preparation for children scheduled for dental treatment under general anesthesia. The findings suggested that such interventions are not effective in reducing anxiety in these children [39]. This notable lack of anxiety improvement may have been related to the different population. In Huntington’s et al. study, the recruited children were of a younger age group than our study. Moreover, in their study, the participants were to be treated under general anesthesia and therefore expected to have more stress than those being treated in regular dental settings. Additionally, the sight of technical equipment and surgical instruments in the operating room could be a factor in increasing anxiety.

Many parameters might influence the level of anxiety, such as sex, age, family socio-economic status, and previous adverse experiences [2,40,41]. In the present study, females scored considerably higher on the CFSS-DS than males. This finding is consistent with the majority of studies available in this field, which found that females reported more dental fear than males, as measured using the CFSS-DS [42,43,44]. This sex difference could be related to the fact that females in general are more likely to feel comfortable expressing their feelings and admitting their fears compared with males [45]. However, some studies which used the same scale reported no effect of sex on dental fear [46,47,48]. Different findings regarding sex could be explained by different study designs and age groups. Sex influences should be considered in conjunction with other factors, such as local culture and family socioeconomic situations.

In regard to the age of the children, studies showed conflicting findings. The current study showed that age was unrelated to dental anxiety. This agrees with Koenigsberg and Johnson and Ten Berge et al., who found no association between age and dental anxiety [42,49]. This is in contrast to Tickle et al. and Dahlander et al., who reported that younger children had greater anxiety [50,51]. Al-Yateem et al. and Sekhavatpour et al. found that older children were more anxious [20,52]. This controversy can be explained by several factors, such as previous traumatic experiences, the cognitive development of older children, and age-group differences in personality traits along with local culture and emotional expression.

The relationship between dental anxiety and previous dental experience is another controversial issue. The present study found statistically significant associations between dental anxiety and previous dental visits with Venham. Children who had previous dental visits had higher odds of having positive behavior. This could have been because children who had previously visited the dentist were familiar with the environment and process, unlike children who had never visited the dentist, who may have had misconceptions about dental procedures. This finding is consistent with Nicolas et al., who found that children who had previously undergone dental treatment had less dental fear than those who had not, as evaluated on the visual analog scale (DF-VAS). In addition, their behavior according to the modified Venham’s scale was better if they had already experienced a dental visit [53]. However, in the current study, this finding was not significant according to CFSS-DS. This is consistent with El-Housseiny et al. and Ma et al., in which previous dental experiences were unrelated to the level of anxiety as measured using CFSS-DS [10,27,44]. These differences in behavior could be related to the nature of the scales used. The CFSS-DS is a self-reported scale that reflects children’s inner emotions toward dental treatment. The Venham scale reflects children’s behavior in the dental office regardless of their inner emotions.

The effect of parental education seems to be another debatable issue. In the present study, a significant relationship with maternal education was found with VCAS and Frankl. Children whose mothers had bachelor’s degrees showed significantly lower odds of having positive Frankl and Venham scores than children whose mothers had lower levels of education. This finding agrees with studies by Rantavuori et al. and El-Housseiny et al., in which children of highly educated parents were more fearful than children with parents of lower education [54,55]. This might be because highly educated parents tend to be overprotective, and parental overprotection is strongly related to children’s anxiety [56]. Moreover, overprotective parents usually spend more time with their children, striving for perfect parenting, and they do not allow them to interact with strangers; this may result in dependent and attached children who grow up to be fearful of strangers.

Milgrom et al. reported that children whose parents has not had any education beyond high school were more afraid of the dentist [57]. The difference between the present study and Milgrom et al.’s may be attributed to the differences between the sample populations: They only targeted low-income American primary school children. Moreover, all the children in the Milgrom et al. study had emergency needs, which might have played a part in the development of dental anxiety.

There was also an association between parental socio-economic status and children’s dental anxiety according to the Venham scale. In this study, children in low-income families had higher anxiety scores than children in higher-income families. This is consistent with Lin et al., who stated that most low-income children had higher dental anxiety scores than higher-income children [40]. This also confirms the finding of Moore et al., who pointed out that poverty is one of the most significant risk factors for the development of dental anxiety [58]. The link between anxiety and financial class could be explained by the fact that lower-income parents confront more challenges, such as limited access to health resources and lack of dental knowledge, which can in turn result in avoidance of regular dental visits, thereby leading to higher anxiety levels.

However, other studies found no significant correlation between parental socio-economic status and children’s dental anxiety [59,60]. The variations between the present study and these studies could be explained by different study designs, different anxiety scales, and different age groups.

This study does have some limitations. First, some young children (mainly those 6 years of age) faced difficulties in using the self-report questionnaire (CFSS-DS). To overcome this difficulty, children were closely assisted by their parents when completing the questionnaire. Second, only restorative dental treatments were included. The rationale for excluding pulp therapy and extraction was to standardize the procedure across all participants. Finally, this study’s sample consisted of 6–8-year-old children who could speak Arabic, seeking dental treatment at King Saud University; therefore, our results cannot be generalized to other populations.

Despite the above-mentioned limitations, this study also has strengths. To the best of our knowledge, this is the first study to assess the effectiveness of a specially designed dental storybook in reducing dental anxiety and improving behavior among children making multiple dental visits, in the form of an RCT with blinding, which made the study groups more comparable and minimized bias and confounding. Moreover, the use of more than one scale to measure anxiety levels aimed to yield better and more representative results [61,62]. Finally, the favorable outcomes of this study offer interesting options for clinicians to discuss with parents and children in an attempt to decrease children’s dental anxiety.

## 5. Conclusions

Preparing children with dental storybooks before visits seems to be effective for decreasing anxiety and improving behavior during dental treatment.

## Figures and Tables

**Figure 1 children-09-00328-f001:**
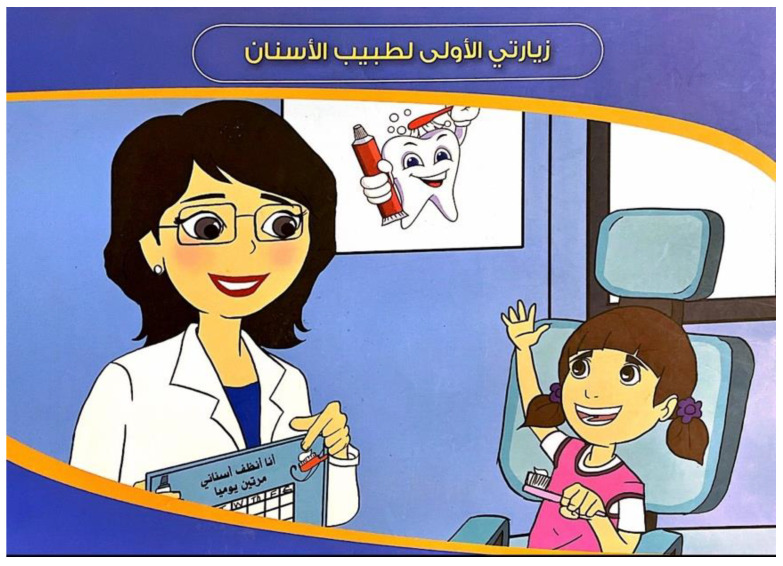
The cover of the book entitled “My First Visit to the Dentist”.

**Figure 2 children-09-00328-f002:**
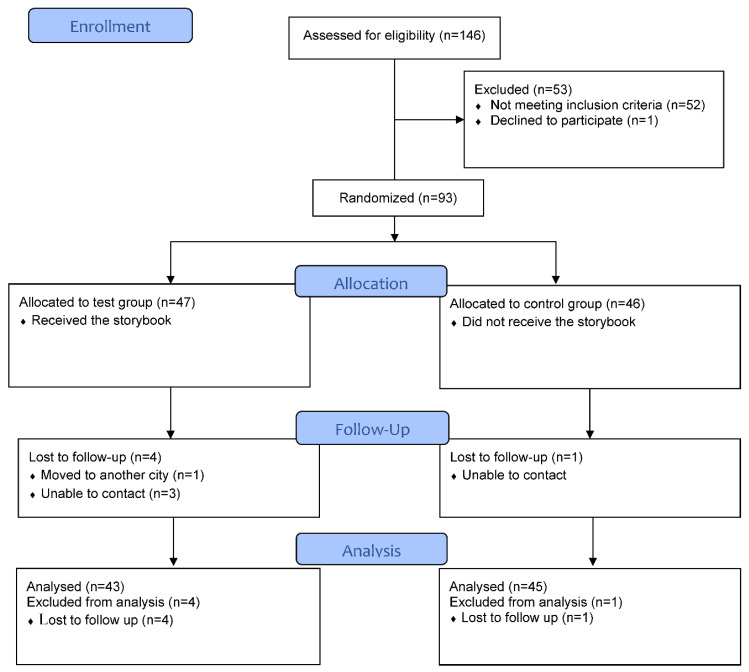
CONSORT flow diagram: the flow of patients during each trial phase.

**Table 1 children-09-00328-t001:** Background characteristics by study groups.

	n	%
Nationality		
Saudi	79	(89.8)
Non-Saudi	9	(10.2)
Sex		
Male	41	(46.6)
Female	47	(53.4)
Age		
6	22	(25)
7	37	(42.1)
8	29	(32.9)
Previous dental visits		
Yes	43	(48.9)
No	45	(51.1)
Family income		
Less than 7700 SAR ^a^	32	(36.4)
7700–22,900 SAR	47	(53.4)
More than 22,900	9	(10.2)
Father education		
High school or less	29	(32.9)
Bachelor	38	(43.2)
High education	21	(23.9)
Mother education		
High school or less	36	(40.9)
Bachelor	42	(47.7)
High education	10	(11.4)
School type		
Governmental	67	(76.1)
Private	21	(23.9)
Relationship to child		
Mother	42	(47.7)
Father	46	(52.3)

^a^ SAR (Saudi Riyal) = 3.75 USD.

**Table 2 children-09-00328-t002:** Comparison of CFSS-DS scores between groups at different visits.

Screening Visit	Mean ± SD ^£^	Median (IQR) ^¥^	*p* Value
Intervention	28.12 ± 8.56	26 (22–35)	0.69 *
Control	28.73 ± 5.75	29 (24–31)	
Examination visit			
Intervention	25.86 ± 8.21	24 (20–31)	0.049 **
Control	29.38 ± 8.93	27 (22–35)	
Treatment visit			
Intervention	23.63 ± 11.36	17 (16–26)	<0.0001 **
Control	33.80 ± 13.35	33 (22–43)	

^£^ Standard deviation (SD); ^¥^ interquartile range (IQR); * calculated using the Student’s *t*-test; ** calculated using Wilcoxon rank-sum.

**Table 3 children-09-00328-t003:** Change in CFSS-DS score between visits in each group.

Intervention	Type of Dental Visits	B Estimate	SE ^£^	*p*-Value
Intervention group	Screening visit	ref	ref	ref
	Examination visit	−0.09	0.05	0.089
	Treatment visit	−0.19	0.05	0.001
Control group	Screening visit	ref	ref	ref
	Examination visit	0.02	0.05	0.726
	Treatment visit	0.15	0.04	0.001

^£^ Standard error (SE).

**Table 4 children-09-00328-t004:** Multivariate mixed-effects negative binomial regression model analysis for factors affecting the patient’s anxiety with CFSS-DS.

Variable	B Estimate	S ^£^	*p* Value
Type of dental visits			0.55
Screening visit	ref	ref	
Examination visit	−0.04	0.04	
Treatment visit	−0.01	0.04	
Study Groups			0.002
Intervention	−0.17	0.06	
Control	ref	ref	
Age			0.70
6	−0.06	0.08	
7	−0.05	0.07	
8	ref	ref	
Sex			0.02
Female	0.14	0.06	
Male	ref	ref	
Previous dental visit			0.14
Yes	0.09	0.06	
No	ref	ref	
Family income			0.46
7700–22,900 SAR	−0.06	0.07	
More than 22,900 SAR	0.05	0.12	
less than 7700 SAR	ref	ref	
Mother Education			0.44
Bachelor degree	0.08	0.06	
High education	0.04	0.11	
High school or less	ref	ref	

^£^ Standard error (SE).

**Table 5 children-09-00328-t005:** Comparison of Venham scores between groups at different visits.

Type of Dental Visits	Behavior	Intervention	Control	*p*-Value *
Screening visit	Positive	26 (44.1)	33 (55.9)	0.20
	Negative	17 (58.6)	12 (41.4)	
Examination visit	Positive	34 (54.8)	28 (45.2)	0.8
	Negative	9 (34.6)	17 (65.4)	
Treatment visit	Positive	34 (66.7)	17 (33.3)	<0.0001
	Negative	9 (24.3)	28 (75.7)	

* Calculated using Chi-squared test.

**Table 6 children-09-00328-t006:** The effects of different visits on the Venham scores of each group.

Intervention	Type of Dental Visits	OR ^£^	95% Confidence Intervals
Intervention group	Screening visit	ref	ref
	Examination visit	2.65	0.95–7.41
	Treatment visit	2.65	0.95–7.41
Control group	Screening visit	ref	ref
	Examination visit	0.47	0.15–1.44
	Treatment visit	0.10	0.03–0.38

^£^ Odds ratio (OR).

**Table 7 children-09-00328-t007:** Multivariate mixed-effects logistic regression model analysis for factors affecting patients’ anxiety with Venham.

Comparison	OR ^£^	95% Confidence Intervals
Type of dental visit		
Screening visit	ref	ref
Examination visit	1.21	0.60–2.46
Treatment visit	0.62	0.31–1.24
Study groups		
Intervention group	2.35	1.21–4.55
Control group	ref	ref
Age		
6	1.03	0.45–2.39
7	1.26	0.59–2.71
8	ref	ref
Sex		
Female	0.86	0.45–1.65
Male	ref	ref
Previous dental visit		
Yes	2.60	1.29–5.23
No	ref	ref
Mother education		
High school or less	ref	ref
Bachelor degree	0.35	0.16–0.77
High education	0.90	0.25–3.19
Family income		
less than 7700 SAR	ref	ref
7700–22,900 SAR	2.65	1.21–5.81
More than 22,900 SAR	2.49	0.67–9.33

^£^: Odds ratio (OR).

**Table 8 children-09-00328-t008:** Comparison of Frankl behavior ratings between groups at different visits.

Type of Dental Visit	Behavior	Intervention	Control	*p*-Value *
Screening visit	Positive	25 (42.4)	34 (57.6)	0.082
	Negative	18 (62.1)	11 (37.9)	
Examination visit	Positive	36 (55.4)	29 (44.6)	0.039
	Negative	7 (30.4)	16 (69.6)	
Treatment visit	Positive	35 (66.0)	18 (34.0)	<0.0001
	Negative	8 (22.9)	27 (77.1)	

* Calculated using Chi-squared test.

**Table 9 children-09-00328-t009:** The effects of different visits on the Frankl scores of each group.

	Type of Dental Visits	OR ^£^	95% Confidence Intervals
Intervention group	Screening visit	ref	ref
	Examination visit	3.79	1.35–10.68
	Treatment visit	3.22	1.18–8.76
Control group	Screening visit	ref	ref
	Examination visit	0.55	0.21–1.47
	Treatment visit	0.18	0.07–0.47

^£^: Odds ratio (OR).

**Table 10 children-09-00328-t010:** Multivariate mixed-effects logistic regression model analysis for factors affecting patient anxiety with Frankl.

	OR ^£^	95% Confidence Intervals
Type of dental visits		
Screening visit	ref	ref
Examination visit	1.48	0.72–3.04
Treatment visit	0.70	0.352–1.39
Study groups		
Intervention group	2.23	1.13–4.39
Control group	ref	ref
Age		
6	1.16	0.49–2.77
7	1.34	0.61–2.96
8	ref	ref
Sex		
Female	0.82	0.42–1.61
Male	ref	ref
Previous dental visit		
Yes	2.01	0.99–4.08
No	ref	ref
Mother education		
High school or less	ref	ref
Bachelor degree	0.36	0.16–0.79
High education	0.80	0.22–2.95
Family income		
less than 7700 SAR	ref	ref
7700–22,900 SAR	2.28	1.03–5.05
More than 22,900 SAR	2.05	0.53–7.91

^£^ Odds ratio (OR).

## Data Availability

The datasets used and analyzed during the current study are available from the corresponding author upon reasonable request.

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
