# Peer review of "The Use of a Dental Storybook as a Dental Anxiety Reduction Medium among Pediatric Patients: A Randomized Controlled Clinical Trial"

_children, 2022, doi:10.3390/children9030328_

Round 1
Reviewer 1 Report
Introduction
-references anxiety and fear in adults several times; as the paper is focused on pediatric patients, I would omit those instances of mentioning adults and rework the opening paragraph (lines 30-37) and other mentions of adults
-include a citation for the statement made in Line 35 about studies referencing anxiety as pain related
-Line 71: I would disagree with the sentiment that using dental stories as a preparatory strategy is controversial. Research is not necessarily anti-social stories. Instead, I would say that previous research has yielded mixed results.
-Lines 81-84 seem run-on and confusing. Please rephrase
-Line 84: Multiple studies have shown the effectiveness on using a storybook as a preparatory tool before dental visits in children with Autism; therefore this sentiment does not hold true as written, and actually emphasizes ableist language (assuming that because no studies have been conducted with neurotypical children means that no studies have been conducted). As a proponent of recognizing and addressing ableist language, I would therefore reframe this sentence to specify that while there have been previous studies looking at the use of dental stories in populations of children with special health care needs like Autism (cite some of those studies), that little/minimal work has been conducted with neurotypically developing children. See https://www.ncbi.nlm.nih.gov/pmc/articles/PMC5447216/ for an example of research with children with Autism in Saudi Arabia (article is by Murshid, doi: 10.15537/smj.2017.5.17398, who is actually one of the authors of the present paper)
Methods
- very clear and well written
-Line 140 and 141 - independent investigator is echoed very close together in text. Consider altering wording
-Section 2.5.2: describe more about the practitioner who conducted the dental exams - their training, their profession, etc.
-Combine lines 187 and 188 about the Frankl scale and reliability - does not need to be 2 sentences
Results
-very impressive results
-echo using "moreover" in line 386 and 388. Consider altering vocabulary
-Statement in lines 395-397 about girls expressing themselves more than boys needs a citation
-Line 428, change controversial
Overall
The study is well conceived and well written. I think once edits are made, it will significantly add to the literature on dental interventions for pediatric populations.
Author Response
Dear reviewer
Please see the the attachment
Thank you
Best regards

Reviewer 2 Report
Title: appropriate
Abstract: line 18, 20, 24: perhaps consider "more cooperative" as a descriptor rather than "better" behavior.
Introduction: Well-written, appropriate scope
M&M: P4, line 148: "A" is italicized.
2.5.3: The "Treatment visit" describes children only requiring occlusal restorations under local anesthesia were selected. I would consider moving this to the inclusion criteria. Additionally, information regarding the standard operating procedure would be valuable (type of local anesthesia, isolation, and restorative material).
Some questions regarding the methodology: Was the operator blinded to the group assignment? Was the operator the outcome assessor? Did the same operator complete all treatment in both groups, and if not, how many operators treated subjects? Were the appointments recorded for assessment, or was the assessor in the operatory for each treatment?
Regarding the story book, I would appreciate a bit more information regarding the content. Were operative procedures and instruments described? Did it show images of local anesthetic, handpieces, etc.? Was it designed for this project or is it available from a publisher? Can a figure be included that shows some content of the book?
In my opinion, this is a well-designed methodology to answer this research question.
Results:
3.2.3: This outcome (the Frankl score for treatment) is significantly superior for the book group and particularly meaningful for clinicians. A pharmacological intervention may not have shown such a large difference. It's personally surprising that a simple book would lead to such a dramatic difference in Frankl scores, although the described methodology makes group concealment likely. I wonder if the authors can include in the discussion the likelihood of the evaluator becoming aware of the group assignment given the child/parent communication during the visit.
Discussion: Appropriate and thorough.
Conclusions: Justified by the results.
Author Response
Dear reviewer
please see the attachment
Thank you and best regards
